# Investigating the Incidence of Dyslipidemia among Brazilian Children and Adolescents Diagnosed with Type 1 Diabetes Mellitus: A Cross-Sectional Study

**DOI:** 10.3390/diseases12030045

**Published:** 2024-02-24

**Authors:** Rafael Fagundes Melo, Lucas Fornari Laurindo, Katia Portero Sloan, Lance Alan Sloan, Adriano Cressoni Araújo, Piero Bitelli, Tereza Laís Menegucci Zutin, Rodrigo Haber Mellen, Luciano Junqueira Mellen, Elen Landgraf Guiguer, João Paulo Cera Albarossi, Márcia Rocha Gabaldi, Patricia Cincotto dos S. Bueno, Jesselina Francisco dos Santos Haber, Sandra Maria Barbalho, Eduardo Federighi Baisi Chagas

**Affiliations:** 1Postgraduate Program in Structural and Functional Interactions in Rehabilitation, School of Medicine, Universidade de Marília (UNIMAR), Marília 17525-902, SP, Brazil; rafael_fagundes_melo@outlook.com (R.F.M.); efbchagas@gmail.com (E.F.B.C.); 2Department of Biochemistry and Pharmacology, School of Medicine, Faculdade de Medicina de Marília (FAMEMA), Marília 17519-030, SP, Brazil; 3Department of Biochemistry and Pharmacology, School of Medicine, Universidade de Marília (UNIMAR), Marília 17525-902, SP, Brazil; 4Texas Institute for Kidney and Endocrine Disorders, Lufkin, TX 75904, USA; kaportero@gmail.com (K.P.S.); tikedlufkin@gmail.com (L.A.S.); 5Department of Internal Medicine, University of Texas Medical Branch, Galveston, TX 77555, USA; 6Department of Animal Sciences, School of Veterinary Medicine, Universidade de Marília (UNIMAR), Marília 17525-902, SP, Brazil; 7Interdisciplinary Center on Diabetes (CENID), Universidade de Marília (UNIMAR), Marília 17525-902, SP, Brazil

**Keywords:** Type 1 Diabetes Mellitus, HbA1c, dyslipidemia, lipoproteins, Brazilian, children, adolescents

## Abstract

The treatment of Type 1 Diabetes Mellitus (T1DM) has always been a challenge for health professionals in relation to glycemic control. Increased body fat has been related to a worsening of the lipid profile and increased prevalence of dyslipidemia in this population, leading to negative repercussions on the control of cardiovascular risk. We aimed to investigate the distribution of lipid levels and the presence of dyslipidemia in children and adolescents with T1DM. A cross-sectional observational study was conducted with 81 individuals of both sexes (4–19 years) diagnosed with T1DM. Anthropometric and biochemical data were collected, in addition to data on physical activity level, sexual maturation stage, and insulin administration regimen. Lipid levels were categorized as normal, borderline, and elevated, and the presence of dyslipidemia was diagnosed by the presence of one or more altered lipid parameter. We noted a prevalence of dyslipidemia in 65.4% of the participants when considering borderline lipid values. Of those, 23.5% had one altered lipid level, and 42.0% had two or more. The main altered lipid levels were total cholesterol and triglycerides, followed by non-HDL-c. The main factor associated with the worsening of lipid levels was the increase in HbA1c. Sex had a significant effect on the levels of TC, HDL-c, and ApoA-I. The results of this study reinforce the need to monitor lipid profile in children and adolescents with T1DM, as well as the importance of early intervention in treating dyslipidemia, especially in patients with poor glycemic control.

## 1. Introduction

The presence of diabetes is one of the main risk factors for cardiovascular diseases (CVDs), which are among the leading causes of death in the world [1,2,3]. In 2021, there were around 8.4 million individuals with Type 1 Diabetes Mellitus (T1DM) in the world, and of these, about 18% were under 20 years old. In that year, about half a million new cases were identified, and about 35,000 non-diagnosed subjects died within twelve months of symptomatic onset. The projection for 2040 is 60–107% higher than in 2021 [4].

T1DM is an autoimmune disease that leads to the destruction of insulin-producing pancreatic ß cells. The highest incidence is observed in childhood and adolescence, which has also been related to a greater risk of complications due to longer exposure time throughout life [5]. It can occur quickly in children and young people with low residual insulin production or slowly in young adults with residual insulin production [6,7,8,9].

The development of T1DM is characterized by the presence of hyperglycemia and clinical symptoms such as polydipsia, polyuria, polydipsia, unexplained weight loss, nocturia, and diabetic ketoacidosis (nausea, vomiting, drowsiness, and coma, the latter of which may lead to death). The laboratory diagnostic criteria consist of fasting plasma glucose greater than or equal to 126 mg/dL, glycemia after the oral glucose tolerance test equal to or greater than 200 mg/dL, or glycated hemoglobin (HbA1c) greater than or equal to 6.5% [10,11,12].

Treatment strategies mainly focus on glycemic control through insulin therapy, which seeks to mimic the action of insulin through the bolus/basal strategy [13]. Even with the emergence of new technologies for continuous blood glucose monitoring, continuous insulin infusion systems, and new types of insulin, a high prevalence rate of complications associated with inadequate glycemic control has been observed [14,15,16].

Screening for risk factors is necessary due to the high risk and number of complications among patients with T1DM. Plasma lipids are considered a relevant risk factor for cardiovascular diseases and are part of monitoring complications in patients with T1DM; therefore, assessing these parameters has been strongly recommended, even in children and adolescents [17]. T1DM patients are particularly susceptible to CVD risk factors, including dyslipidemia, which, over time, contribute to additional heart complications [18,19,20].

Another important factor is the increased prevalence of overweight and obesity in patients with T1DM, which is associated with an increased risk of the early development of dyslipidemia and cardiovascular diseases [21]. Overweight and obesity in T1DM are also related to elevated HbA1c values and a shorter lifespan, which, in turn, increases the risk of microvascular and macrovascular complications [22].

Epidemiological studies and clinical trials have used total cholesterol and its traditional fractions to identify the prevalence of dyslipidemia. However, there is growing concern about changes in the concentrations of the apolipoproteins (Apo)B and ApoA-I in children and adolescents with T1DM. Monitoring these apolipoproteins is mainly useful in monitoring cardiovascular risk of T1DM patients who have low-density lipoprotein cholesterol-(LDL-c) and high-level lipoprotein-cholesterol (HDL-c) values within the normal range [23,24].

Although LDL-C continues to be the main target in screening for dyslipidemia in clinical practice in children and adults [25], non-HDL-C cholesterol has been suggested as a useful measure in children and for more accurate assessments of the risk of atherosclerotic cardiovascular disease in adults [26]. Due to the increasing incidence of abnormal lipid levels in the young population with T1DM, lipid level screening can be crucial for preventing future atherosclerotic cardiovascular diseases. For these reasons, we investigated the distribution of lipid levels and the presence of dyslipidemia in children and adolescents with T1DM.

## 2. Materials and Methods

### 2.1. Study Design

This was a cross-sectional observational prevalence study. Patient data were obtained during routine consultations at the Specialty Medical Outpatient Clinic of the University of Marília between the years of 2019 and 2020 and stored in the Interdisciplinary Center for Diabetes (CENID) of the Universidade de Marília (UNIMAR)—SP, Brazil database. This study is part of a research project previously approved by the UNIMAR Ethics and Research Committee with protocol number 3.606.397/2019 (CAAE: 20492619.6.0000.5496).

### 2.2. Study Population

The sample size was calculated using G*Power software, version 3.1.9.2 (Franz Faul, UniversitätKiel, Germany), to estimate the high ApoB prevalence in children and adolescents with T1DM. Considering an expected proportion of 10% (0.10) and a medium effect size (0.15) [27], a minimum sample size of 79 sample elements was estimated for a type I margin of error (α) of 5% and a study power of 95%.

Considering the eligibility criteria, 81 patients of both sexes (male = 48; female = 33), aged between 4 and 19 years old, diagnosed with T1DM for at least twelve months and with C-peptide values < 0.3 ng/mL that signed the Informed Consent Form were included in the study. It is worth highlighting that the patients in our study were lipid-lowering-therapy naive. Patients with a diagnosis of Autism Spectrum Disorder, physical or mental disability with compromised self-care, and incomplete lipid level data were excluded from the study.

### 2.3. Study Variables

Sociodemographic characteristics (age, sex), anthropometric and biochemical data, and data on physical activity level (PAL), sexual maturation stage, and insulin administration regimen were collected for all participants. The presence of comorbidities was assessed to characterize the sample and control confounding variables. Data on comorbidities were recorded as present or absent without identifying the condition.

Anthropometric measurements of waist circumference (cm), body mass (kg), and height (meters) were taken. Body mass and height were used to calculate the body mass index z-score (BMI-z), and the subjects were categorized as underweight, normal weight, overweight, and obese according to the recommendations of the World Health Organization [28]. Their fat and lean mass percentages were estimated by bioimpedance testing, and the fat values were categorized as obese for values > 30% for girls and >25% for boys (Qian et al., 2020). Waist circumference, body weight, and height measurements were used to calculate the conicity index to analyze visceral fat [29,30].

Habitual physical activity pattern was assessed using the physical activity recall proposed by Bouchard [31]. One week (seven days) was recorded to estimate daily energy expenditure, expressed in kcal per kilogram of body weight per day (kcal/kg/day). The resting metabolic rate (RMR), expressed in kcal/day, was estimated using an age- and sex-specific equation [32,33]. The equation used to determine physical activity level (PAL) includes daily energy expenditure (kcal/day) divided by resting metabolic rate (kcal/day). PAL values were classified as mild (women < 1.56; men < 1.55), moderate (women 1.56 to 1.82; men 1.55 to 2.10), and vigorous (women > 1.82; men > 2.10) [34].

Laboratory tests were carried out on fasting blood glucose, HbA1c, total cholesterol (TC), LDL-c, HDL-c, triacylglycerides (TG), Apoliprotein-A1 (ApoA), and Apoliprotein-B (ApoB). Glycemic control was assessed by fasting blood glucose and glycated hemoglobin (HbA1c). Normal values for fasting blood glucose were considered to be <100 mg/dL. HbA1c values were categorized as adequate (<7%) and not adequate (≥7%) [35,36]; however, due to the characteristics of the population of the study, HbA1c values were also categorized as less than 7%, from 7 to 8%, and greater than 8% [8]. LDL-c was calculated using the Friedewald equation [37], and non-HDL-c was calculated using the CT—HDL-c equation. For diagnostic interpretation, lipid parameters were categorized as normal, borderline, and elevated [38]. However, in the Brazilian population, it is recommended to use borderline values with a cut-off point to identify altered values, and the presence of dyslipidemia is defined by the presence of at least one altered lipid parameter considering TC ≥ 170 mg/dL; LDL-c ≥ 110 mg/dL; HDL-c ≤ 45 mg/dL; non-HDL-c ≥ 120 mg/dL; TG ≥ 75 mg/dL (0 to 9 years) or ≥90 mg/dL (10 to 19 years); and ApoA-I < 120 mg/dL and ApoB ≥ 90 mg/dL (SBP Scientific Department of Endocrinology, 2020). In addition, to allow comparison with other studies lipid parameters were also classified as elevated when TC ≥ 200 mg/dL; LDL-c ≥ 130 mg/dL; HDL-c < 40 mg/dL; non-HDL-c ≥ 145 mg/dL; TG ≥ 100 mg/dL (0 to 9 years) and ≥ 130 mg/dL (10 to 19 years); ApoA-I < 115 mg/dL and ApoB ≥ 110 mg/dL [39,40,41]. The number of lipid parameters altered for both cut-off points was considered to analyze the results.

Although there are normative values for age and sex in relation to percentile distribution, the diagnosis of dyslipidemia is based on cut-off points that classify lipid parameters as “acceptable (normal)”, “borderline”, and “elevated/high” [42]. It has been documented that there is a wide variation in the cut-off points used to identify the prevalence of changes in TC (borderline ≥ 170 and high ≥ 200 md/dL) and LDL-c (borderline ≥ 110 and high ≥130 md/dL) in children and adolescents [43]. The differences between studies regarding prevalence distributions may be related to the cut-off points adopted, and for this reason, the results were presented considering both borderline and elevated values.

Data on the insulin administration schedule were obtained considering information on total insulin (U/day), bolus insulin (U/day), and basal insulin (U/day). The values for the insulin strategy were converted into units (U)/kg/day. Considering the total insulin (U/kg/day), the administration schedule was classified as “below recommended for weight”, “recommended for weight”, and “above recommended for weight” according to the duration of the disease and sexual maturation stage [16]. Patients were divided into two groups regarding the method of insulin administration: patients using a continuous insulin infusion system (SICI—insulin infusion pump) and patients using multiple doses of insulin (MDI).

Regarding the time of diagnosis in years, the data were categorized into less than five years (<5 years) and greater than or equal to five years (≥five years), considering the increase in cardiovascular risk with time of exposure to the disease (Bjornstad et al., 2018; Cortez et al., 2015). Sexual maturation stage was assessed using the Tanner sexual maturation scale and categorized into pre-pubertal, pubertal, and post-pubertal [44].

The Strengthening the Reporting of Observational Studies in Epidemiology (STROBE) statement was followed in this study.

### 2.4. Data Analysis

Quantitative variables are presented as mean value and 95% confidence interval (95% CI). Qualitative variables are described as absolute and relative frequency distribution. The 95% CI was calculated using the Bootstrap technique for percentile and the resampling of 1000 sample elements. For the analysis of the 95% CI, significant differences were considered in the absence of an intersection between the lower and upper limits of the 95% CI. The association between qualitative variables was analyzed using the Chi-square test. The assumption of homogeneity of variances was verified using the Levene test to compare the means. To analyze the differences in means for two independent groups, Student’s *t*-test was performed for unpaired samples. For analyses comparing the means of more than two independent groups, a one-way ANOVA was performed, followed by a post hoc Least Significant Difference test when necessary. Multiple linear regression analyses explored the influencing factors on lipid parameters. The selection of independent variables for the multiple linear regression analyses was based on the physiological assumption of lipid metabolism. Multiple linear regression models were built using the Backward method, and linear R^2^ was used to estimate the percentage of variation in the dependent variable explained by the variation in the independent variables inserted in the model. Only the final models with the best fit to the data were presented for the regression analysis results. The significance level adopted was 5%, and the data were analyzed using SPSS software version 27.0.

## 3. Results

Table 1 presents the characteristics of the population. The sample consisted of 81 children and adolescents of both sexes (59.3% male and 40.7% female) diagnosed with T1DM aged between 4 and 19 years old; 72.8% used the MDI, 75.3% had HbA1c values greater than 7%, and 71.6% had an insulin administration schedule appropriate for their weight. Although 18.5% had an insulin administration schedule (U/kg) below recommended, no significant difference was observed between the values of plasma glucose and HbA1c (%) among the insulin administration schedule categories (U/kg) (Figure 1).

We did not observe a relationship between the insulin administration schedule and nutritional status (BMI z-score), duration of diabetes, insulin administration method, and PAL. Likewise, no significant differences were found between patients with different insulin administration schedules when comparing the mean BMI z-score, percentage of fat, percentage of lean mass, PAL score, and time since diagnosis. Age was found to be a significant factor among patients with an adequate or above-expected schedule for their weight (results not shown). It is important to note that all patients with an insulin administration schedule lower than expected for their weight were in the post-pubertal stage.

Regarding sexual maturation, a similar distribution was observed between the stages. Most of the individuals did not present associated comorbidities and were classified as low active by PAL score. According to the BMI z-score, 24.7% of the individuals were overweight and obese. A similar proportion of individuals were classified as obese by the percentage of body fat (23.5%) (Table 1).

Table 2 presents the distribution of lipid levels and the prevalence of dyslipidemia in the population studied. For borderline values, TC was the lipid parameter with the highest prevalence of changes among patients (42.0%) followed by non-HDL-c (35.8%) and TG (33.3%). The great majority of the individuals presented normal values of ApoA-I (95.0%).

Considering the presence of at least one altered lipid parameter as a diagnostic criterion for dyslipidemia, 65.4% of the individuals presented the disease; however, when considering high cut-off points as the diagnostic criterion, the prevalence decreased to 32.1%. When looking at borderline lipid level parameters, 23.5% presented one altered parameter and 42.0% presented two or more (Figure 2A), and when considering high values, 13.6% had one altered lipid parameter, and 18.5% had two or more (Figure 2B).

It was observed that patients with HbA1c values < 7% had higher HDL-c values than those with HbA1c values between 7 and 7.9% (*p* < 0.05). The same trend was observed for ApoA-I (Table 3).

It was noted that patients with HbA1c < 7% had a lower prevalence of dyslipidemia, as well as changes in lipid parameters of TC, non-HDL-c, and ApoA-I. Among patients with HbA1c < 7% and ≥8%, a lower proportion of LDL-c changes was observed. In patients with HbA1c < 7%, a higher proportion of altered values of TG, HDL-c, and ApoB was noted in comparison with other HbA1c categories (Table 4).

A multiple linear regression analysis was performed to identify the independent variables that significantly affect the variation in lipid parameters. For this analysis, the independent variables of age, sex, time of diagnosis, BMI z-score, percentage of fat, percentage of lean mass, conicity index, pubertal stage, HbA1c, and insulin administration schedule were considered. It was observed that an increase in HbA1c and the female sex contributed to the increase in TC. Increased age, duration of disease, HbA1c, sexual maturation stage, and insulin administration schedule contributed to the increase in serum TG values. The increase in HbA1c had a significant effect on increasing values of LDL-c, non-HDL-c, and ApoB. Female sex had a significant impact on the increasing values for HDL-c and ApoA-I (Table 5).

## 4. Discussion

The results of our study show a prevalence of dyslipidemia in children and adolescents with T1DM of 65.4% when borderline lipid values are considered and 32.1% when high lipid values are considered as a diagnostic criterion. The main altered lipid parameters were TC and TG, followed by non-HDL-c, and the foremost factor associated with the worsening of lipid parameters was the increase in HbA1c. Sex had a significant effect on HDL-c and ApoA-I parameters, and in addition to HbA1c, increases in age, duration of disease, sexual maturation stage, and insulin units per kg were related to increased values of TG.

In a study carried out with children and adolescents with DM1, high non-HDL-c values (>120 mg/dL) were found in 30% of the sample, and increased HbA1c, time since diagnosis, and being female were related to an increase in non-HDL-c values [45]. Although our results showed similar values regarding prevalence, only HbA1c had a significant effect on non-HDL-c. In addition to being an easily accessible and recommended measure for diagnosing dyslipidemia [43], elevated non-HDL-c values can provide additional prognostic information in low-LDC-c conditions [26].

Although CVD rarely manifests during childhood, subclinical damage to the cardiovascular system begins to develop from an early age, and prevention through reducing exposure to cardiovascular risk factors, in particular maintaining normal values of lipid levels, is one of the main medical therapy targets for the treatment of T1DM [46]. It is well known that the risk of cardiovascular diseases is increased among people with diabetes, mainly in those with poor glycemic control [47,48,49]. As a result, the guidelines suggest aggressive goals for lipid levels, especially in relation to LDL-c. The increased CVD risk during childhood in people with T1DM and DM2 is related to prolonged exposure to hyperglycemia, which has an oxidative effect that leads to the formation of advantageous glucose end-products [50,51,52].

In the general population of children and adolescents, the prevalence of dyslipidemia is 20%, regardless the presence of comorbidities [53]. The prevalence of dyslipidemia in children and adolescents with Type 2 Diabetes Mellitus documented in the literature ranges from 67.5% [54] to 47.2% [55].

The high prevalence of dyslipidemia in children and adolescents with T1DM found in our study (65.4%) is similar to the ones documented in recent studies (67.3%) that analyzed the same population and used the same diagnostic criteria, regardless of the cut-off points adopted and the clinical aspects of glycemic control, sexual maturation stage, sex, PAL, and body composition [42,43,56]. When using elevated cut-off points as diagnostic criteria for dyslipidemia, the prevalence found was 47.2% [57] and 49.5% [58].

Several studies with individuals with T1DM have evaluated the relationship between the prevalence of dyslipidemia and the duration of the disease. A prevalence of 37.1% was found in patients with up to five years of diagnosis and 53.6% in patients with up to ten years of diagnosis of T1DM [59]. Other studies have shown a prevalence of 26.2% in patients diagnosed with T1DM for 5.6 years [60] and 72.5% in patients with the disease for 10.6 years [61].

The main factor associated with worsening lipid parameters was the increase in HbA1c, except for HDL-c and ApoA-I. An increase in TG, LDL-c, and non-HDL-c values has been observed in patients with poor glycemic control. This may be related to an inadequate insulin administration schedule, as insulin has an antilipolytic effect by inhibiting lipase, an insulin-sensitive hormone in adipose tissue, which reduces the secretion of free fatty acids [62].

Our results show a close relationship between high HbA1c values and altered ApoA-I. Other studies are consistent with these results and show that the biomarkers of hyperglycemia, hypertriglyceridemia, higher ApoA-I, and ApoB–to–ApoA-I ratio are significantly associated with T1DM risk [40,63,64,65].

Although an effect of the insulin administration schedule on TG values was observed in our study, it is worth noting that no association was observed between the insulin administration schedule and glycemic control evaluated by HbA1c. The literature illustrates that factors such as body composition, diet, physical activity, and sexual maturation stage may contribute to an inadequate insulin administration schedule [66,67].

Sex was an important factor associated with total cholesterol, HDL-c, and ApoA-I, with higher values among females. A higher prevalence of dyslipidemia has been observed in females, which has been related to a higher prevalence of overweight and obesity and which has demonstrated an effect on increasing TG and LDL-c, as well as reducing HDL-c [68]. In children and adolescents with T1DM, females have been shown to suffer a greater impact from the disease and an increased presence of cardiovascular risk factors [69].

The increased prevalence of overweight and obesity in patients with T1DM is associated with worsening of the lipid profile and the increased prevalence of dyslipidemia in this population [21]. Furthermore, there is evidence that HDL-c does not fulfill its cardioprotective function in T1DM due to its dysfunctional form [70]. Therefore, the lower prevalence of changes in HDL-c and ApoA-I observed in our study does not guarantee a protective factor in relation to the high prevalence of changes in TC, TG, and non-HDL-c, although LDL-c and ApoB showed lower proportions of altered values.

It has been suggested that although HDL-c and ApoA-I are considered atheroprotective, in prooxidant and inflammatory conditions such as diabetes and obesity, chemical modifications such as oxidation and nitration can result in dysfunctionality and abnormality, promoting increased cardiovascular risk. Even with the possible functional changes in HDL-c and ApoA-I in pathological conditions that may generate controversy about their use in diagnosing dyslipidemia, their use has been recommended [71].

Despite the better glycemic control of patients using SICI, the insulin administration method showed no significant effect on the variation in lipid parameters. Significant effects of increasing age, duration of diabetes, HbA1c, sexual maturation stage, and insulin administration schedule (insulin/kg) were observed on the increase in TG values. However, in a large cohort study, a better lipid profile was observed in patients using SICI compared to MDI, as well as associations of increased lipids in females and with increasing age, duration of diabetes, HbA1c, and BMI [72].

The use of all lipid parameters recommended for dyslipidemia screening contributes to increasing external validity in relation to prevalence estimates. Many studies that have carried out surveys of the prevalence of dyslipidemia in the population of children and adolescents with T1DM did not carry out estimates based on the dosage of all suggested lipid parameters, and this could lead to prevalence estimates that are lower than reality. In Brazil, due to economic issues, the dosage of Apo A and B is not very frequent, which may be contributing to estimates of the prevalence of dyslipidemia lower than that observed in the present study.

One of the limitations of this study is the fact that it was not possible to investigate the presence of familial hypercholesterolemia. Regardless, our results are very similar to the findings in the literature in respect of the prevalence of dyslipidemia and associated factors. Given the impact of changes in lipid parameters and dyslipidemia on cardiovascular risk on individuals with T1DM, pharmacological treatment has been recommended in addition to lifestyle interventions. Statins represent the first-line pharmacological option for treating dyslipidemia in children and adolescents despite the rare potential side effects presented by this therapy.

Although physical exercise and diet are widely recommended for changes in lipid parameters other than LDL-c, in conditions where lifestyle modifications are not capable of producing significant improvement, the study of statins is recommended. It is worth highlighting that changes in lifestyle are part of the recommendations for the treatment of DM1, but an increase in the prevalence of a sedentary lifestyle among children and adolescents has been observed [42].

Other agents are still being studied in children for long-term efficacy, safety, and tolerability [72]. Although this cross-sectional observational study has its limitations, regarding the cause and effect relationship, the results of the multiple linear regression analysis provide important insights for directing clinical practice for the interdisciplinary care team when constructing more robust cohort studies.

## 5. Conclusions

A high prevalence of dyslipidemia was found in children and adolescents with T1DM, with the main altered lipid parameters being total cholesterol and TG, followed by non-HDL-c. Despite the low prevalence of altered LDL-c and ApoB values, these lipid parameters are of great relevance in clinical practice due to their great impact as a risk factor for cardiovascular diseases. The lack of glycemic control, assessed by HbA1c, was the main factor associated with the worsening of lipid parameters. The results of the present study reinforce the need to monitor the lipid profile of children and adolescents with T1DM, as well as the importance of early intervention in treating dyslipidemia, especially in patients with difficulties in adhering to treatment and achieving adequate glycemic control.

## Figures and Tables

**Figure 1 diseases-12-00045-f001:**
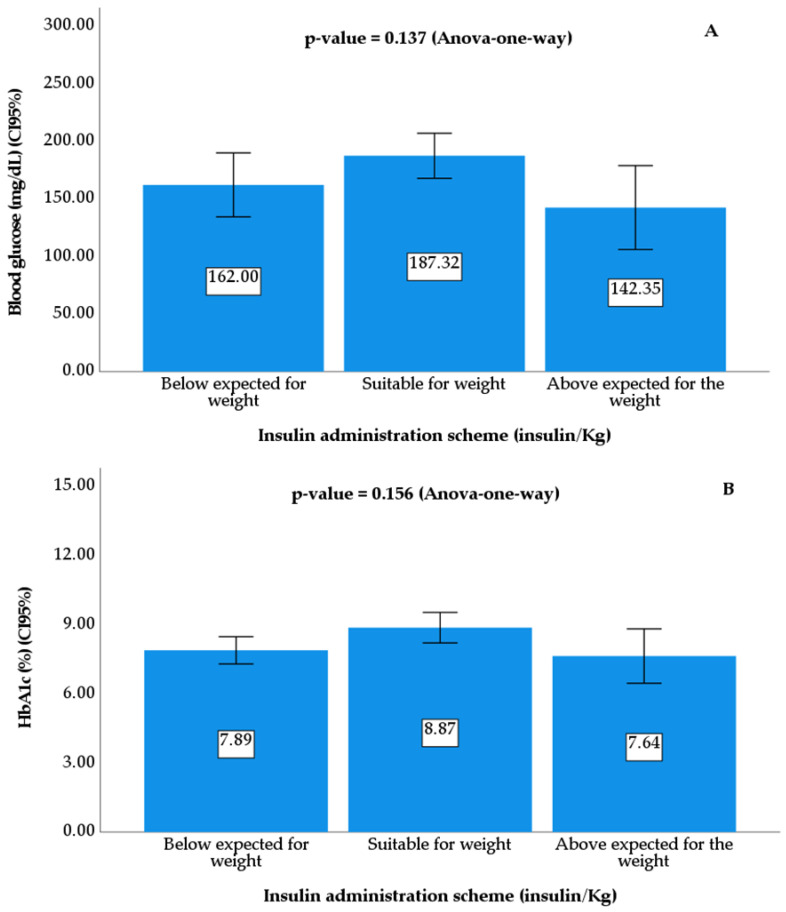
Comparison of the means and 95% confidence intervals (error bars) of mean glycemia (**A**) and HbA1c (**B**) between patients with an insulin administration schedule below, adequate, and above the expected body weight. Note: The *p*-value was calculated using a one-way ANOVA.

**Figure 2 diseases-12-00045-f002:**
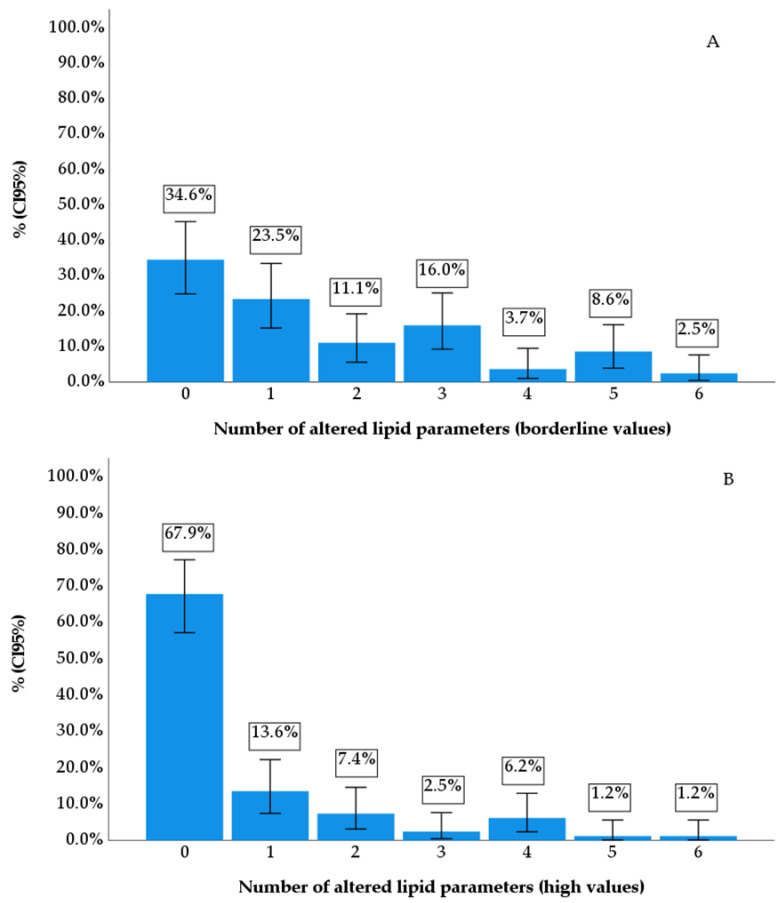
Relative frequency distribution (%) with 95% confidence interval (95% CI) using the Bootstrap technique for the number of altered lipid parameters considering borderline (**A**) and elevated (**B**) values.

**Table 1 diseases-12-00045-t001:** Absolute (f) and relative (%) frequency distribution of sample characteristics.

Parameters		*f*	%	CI95%	
				LL	UL
Gender	Male	48	59.3	48.1	70.3
	Female	33	40.7	29.7	51.9
Diagnostic time	<5 years	46	56.8	45.7	66.7
	>5 years	35	43.2	33.3	54.3
Pubertal staging	Pre-pubescent	21	25.9	17.3	35.8
	Pubescent	28	34.6	24.7	45.7
	Post-pubertal	32	39.5	28.4	50.6
Associated comorbidities	Yes	5	6.2	1.2	11.1
	No	76	93.8	88.9	98.8
PAL	Low active	70	86.4	77.8	93.8
	Moderate active	11	13.6	6.2	22.2
Insulin administration	CIIS	22	27.2	18.5	37.0
	MDI	59	72.8	63.0	81.5
Glycated hemoglobin	<7%	20	24.4	14.6	34.1
	7 a 7.9%	15	18.3	11.0	26.8
	≥8%	47	57.3	46.3	67.1
Insulin/kg	Below expected for weight	15	18.5	9.9	27.2
	Adequate for weight	58	71.6	60.5	81.5
	Above expected for the weight	8	9.9	3.7	17.3
MI-z nutritional status	Underweight	9	11.1	4.9	18.5
	Normal weight	52	64.2	53.1	75.3
	Overweight	18	22.2	13.6	32.1
	Obese	2	2.5	0.0	6.2
Obesity (fat %)	Obese	19	23.5	14.8	32.1
	Not obese	62	76.5	67.9	85.2

Note: 95% confidence interval (95% CI); lower limit (LL); upper limit (UL); multiple doses of insulin (MDI); continuous insulin infusion system (CIIS); physical activity level (PAL); body mass index z-score (BMI-z).

**Table 2 diseases-12-00045-t002:** Absolute (f) and relative (%) frequency distribution with 95% confidence interval (95% CI) using the Bootstrap technique for changes in lipid parameters and presence of dyslipidemia considering borderline and high values as cut-off points.

Lipid Profile		*f*	%	CI95%	
				LL	UL
TC	Borderline	34	42.0	30.9	51.9
	Elevated	14	17.3	9.9	25.9
TG	Borderline	27	33.3	23.5	44.4
	Elevated	13	16.0	8.6	24.7
LDL-c	Borderline	13	16.0	8.6	24.7
	Elevated	7	8.6	3.7	14.8
HDL-c	Borderline	14	17.3	9.9	25.9
	Elevated	5	6.2	1.2	12.3
non-HDL-c	Borderline	29	35.8	25.9	46.9
	Elevated	13	16.0	8.6	23.5
ApoA-1	Borderline	2	2.5	0.0	6.2
	Elevated	2	2.5	0.0	6.2
ApoB	Borderline	16	19.8	12.3	29.6
	Elevated	6	7.4	2.5	13.6
Dyslipidemia	Borderline	53	65.4	54.3	75.3
	Elevated	26	32.1	22.3	42.0

Note: 95% confidence interval (95% CI); lower limit (LI); upper limit (LS); total cholesterol (TC); triacylglycerides (TG); low-density lipoprotein-cholesterol (LDL-c); high-density lipoprotein-cholesterol (HDL-c); Apolipoprotein B (ApoB); Apolipoprotein A-I (ApoA-I).

**Table 3 diseases-12-00045-t003:** Comparison of the means and 95% confidence intervals (95% CIs) between HbA1c categories.

Parameters	HbA1c									*p*-Value
	<7% (n = 20)			7 a 7.9% (n = 15)			≥8% (n = 46)			
	Mean	CI 95%		Mean	CI 95%		Mean	CI 95%		
		LL	UL		LL	UL		LL	UL	
Age (years)	12.8	11.0	14.5	14.1	12.7	15.5	12.1	11.0	13.2	0.159
Diagnosis time (year)	4.8	3.6	6.0	4.9	3.0	6.7	3.9	3.0	4.8	0.415
PAL (score)	1.20	1.10	1.30	1.30	1.18	1.41	1.24	1.18	1.30	0.366
BMI (z-score)	0.67	0.10	1.23	0.76	−0.05	1.56	−0.09	−0.41	0.23	0.014
Conicity index	1.14	1.09	1.18	1.15	1.10	1.21	1.14	1.12	1.16	0.789
Fat (%)	21.3	17.1	25.4	25.6	21.6	29.5	20.8	18.6	22.9	0.108
Lean mass (%)	78.9	75.0	82.8	76.4	71.9	80.9	78.5	76.3	80.7	0.604
TC (mg/dL)	177.5	165.3	189.7	152.2	136.2	168.1	164.2	153.3	175.2	0.084
TG (mg/dL)	90.9	67.2	114.6	91.3	50.4	132.3	76.1	62.7	89.4	0.449
LDL-c (mg/dL)	93.1	83.6	102.7	79.8	71.5	88.2	90.8	81.2	100.5	0.316
HDL-c(mg/dL)	59.7	53.5	65.8	51.1	44.5	57.8	54.4	51.9	56.8	0.049 *
Não-HDL-c (mg/dL)	117.8	104.5	131.1	101.0	86.7	115.3	109.9	98.3	121.4	0.363
ApoA (mg/dL)	158.3	149.0	167.7	140.0	129.9	150.0	147.9	143.4	152.4	0.007 *
ApoB (mg/dL)	78.5	71.3	85.7	70.3	61.0	79.5	78.9	72.9	84.9	0.282

Note: Lower limit (LL). Upper limit (UL). * indicates a significant difference between means according to a one-way ANOVA for *p*-value ≤ 0.050. Different superscript letters indicate a significant difference between the means according to the post hoc Least Significant Difference test for *p*-value ≤ 0.050. Physical activity level (PAL); body mass index (BMI); total cholesterol (TC); triacylglycerides (TG); low-density lipoprotein-cholesterol (LDL-c); high-density lipoprotein-cholesterol (HDL-c); Apolipoprotein B (ApoB); Apolipoprotein A-I (ApoA-I).

**Table 4 diseases-12-00045-t004:** Absolute (f) and relative (%) frequency distribution with 95% confidence interval of the presence of lipid changes and dyslipidemia by HbA1c category.

Parameters	HbA1c											
	<7% (n = 20)				7 a 7.9% (n = 15)				≥8% (n = 46)			
	*f*	%	CI95%		*f*	%	CI95%		*f*	%	CI95%	
			LL	UL			LL	UL			LL	UL
CT >170 mg/dL	6	30.0	10.0	50.0	7	46.7	20.0	73.3	21	45.7	30.4	60.9
TG >75 ou >90 mg/dL	7	35.0	15.0	55.0	6	40.0	13.3	66.7	14	30.4	17.4	43.5
LDL-c >110 mg/dL	2	10.0	0.0	25.0	6	40.0	13.3	66.7	5	10.9	2.2	21.7
HDL-c <45 mg/dL	5	25.0	10.0	45.0	2	13.3	0.0	33.3	7	15.2	4.4	26.1
Non-HDL-c >120 mg/dL	6	30.0	10.0	50.0	6	40.0	13.3	66.7	17	37.0	23.9	52.1
ApoB >90 mg/dL	6	30.0	10.0	50.0	4	26.7	6.7	46.7	6	13.0	4.3	23.9
ApoA-I<120 mg/dL	0	0.0	0.0	0.0	1	6.7	0.0	20.0	1	2.2	0.0	6.5
Dyslipidemia	12	60.0	35.0	80.0	11	73.3	46.7	93.3	30	65.2	52.2	78.3

Note: 95% confidence intervals (95% CIs) were calculated using the Bootstrap technique. Lower limit (LI). Upper limit (LS). The *p*-value was calculated by Student’s *t*-test for independent samples. Total cholesterol (TC); triacylglycerides (TG); low-density lipoprotein-cholesterol (LDL-c); high-density lipoprotein-cholesterol (H DL-c); Apolipoprotein B (ApoB); Apolipoprotein A-I (ApoA-I).

**Table 5 diseases-12-00045-t005:** Multiple linear regression analysis for the effect of independent variables on total cholesterol and fractions, such as the number of altered lipid parameters.

Parameters		B	CI95% (B)		*p*-Value	Model	
Dependent	Independent		LL	UL		*p*-Value	R^2^
CT (mg/dL)	(Constant)	92.24	59.82	124.66	<0.001 *	<0.001 ⱡ	0.216
	Gender	14.92	1.27	28.57	0.033 *		
	HbA1c (%)	6.07	3.10	9.05	<0.001 *		
TG (mg/dL)	(Constant)	−58.15	−138.02	21.72	0.151	<0.001 ⱡ	0.364
	Age (years)	−6.95	−13.88	−0.03	0.049 *		
	Diagnostic time (years)	7.84	3.94	11.73	<0.001 *		
	HbA1c (%)	6.65	2.28	11.02	0.003 *		
	Pubertal staging	41.41	9.87	72.96	0.011 *		
	Insulin/kg	25.69	1.55	49.83	0.037 *		
LDL-c (mg/dL)	(Constant)	44.91	23.25	66.56	<0.001 *	<0.001 ⱡ	0.184
	HbA1c (%)	5.19	2.75	7.64	<0.001 *		
HDL-c (mg/dL)	(Constant)	48.03	41.03	55.03	<0.001 *	0.037 ⱡ	0.054
	Gender	5.00	0.30	9.70	0.037 *		
Non-HDL-C (mg/dL)	(Constant)	55.28	27.97	82.58	<0.001 *	<0.001 ⱡ	0.178
	HbA1c (%)	6.41	3.33	9.50	<0.001 *		
ApoA-I (mg/dL)	(Constant)	131.23	119.91	142.55	<0.001 *	0.001 ⱡ	0.122
	Gender	12.63	5.04	20.22	0.001 *		
ApoB (mg/dL)	(Constant)	48.99	34.08	63.90	<0.001 *	<0.001 ⱡ	0.161
	HbA1c (%)	3.29	1.61	4.98	<0.001 *		

Note: Gender (1 = male; 2 = female). Pubertal staging (1 = pre-pubertal; 2 = pubescent; 3 = post-pubertal). Insulin/kg administration schedule (1 = below expected for weight; 2 = adequate for weight; 3 = above expected for weight). Intercept (Constant). Regression coefficient (B). 95% confidence interval (95% CI). Lower limit (LI). Upper limit (LS). * indicates a significant effect of the independent variable for *p*-value ≤ 0.050. ⱡ indicates a significant effect of the model for *p*-value ≤ 0.050. Linear R^2^ (estimate of the percentage of variation in the dependent variable explained by the variation in the independent variables included in the model). *p*-value ≤ 0.050 indicates a significant effect. Total cholesterol (TC); triacylglycerides (TG); low-density lipoprotein-cholesterol (LDL-c); high-density lipoprotein-cholesterol (H DL-c); Apolipoprotein B (ApoB); Apolipoprotein A-I (ApoA-I).

## Data Availability

Data are contained within the article.

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
