# Peer review of "Investigating the Incidence of Dyslipidemia among Brazilian Children and Adolescents Diagnosed with Type 1 Diabetes Mellitus: A Cross-Sectional Study"

_diseases, 2024, doi:10.3390/diseases12030045_

Round 1
Reviewer 1 Report
Comments and Suggestions for Authors
The authors cross-sectionally examined the incidence of dyslipidaemia among children and adolescents diagnosed with type 1 diabetes mellitus (T1DM) in the Marilia region (Brazil). The paper is well organized and very concisely written. Some important issues must be addressed to qualify the paper for publishing.
Main obstacle (potentially disqualifying):
1. the authors mentioned that the participants have been diagnosed with T1DM for at least twelve months and are hypolipaemic treatment-naive. What is the reason why the lipid status is not assessed at the moment of diagnosing T1DM? In those circumstances, the results you presented are not correct, as you have observed the (expectedly insufficient) effects of applied insulin treatment on the lipid status of a certain patient. Or, on the other hand, why did the mentioned patients not get hypolipaemic drugs?
Minor obstacles:
1. The introduction must be improved with a re-selection of data about diabetes (too many data points are not linked to dyslipidaemia).
2. Table 1. Multiple doses of insulin are abbreviated as MDI, not as MID.
3. Gentle English polishing is needed.
Comments on the Quality of English Language
1. Gentle English polishing is needed.
2. Table 1. Multiple doses of insulin are abbreviated as MDI, not as MID.
Author Response
The authors cross-sectionally examined the incidence of dyslipidaemia among children and adolescents diagnosed with type 1 diabetes mellitus (T1DM) in the Marilia region (Brazil). The paper is well organized and very concisely written. Some important issues must be addressed to qualify the paper for publishing.
Response: Dear doctor, thank you for you time reviewing this manuscript. We are very glad to correct according to your suggestions.
Main obstacle (potentially disqualifying):
The authors mentioned that the participants have been diagnosed with T1DM for at least twelve months and are hypolipaemic treatment-naive. What is the reason why the lipid status is not assessed at the moment of diagnosing T1DM? In those circumstances, the results you presented are not correct, as you have observed the (expectedly insufficient) effects of applied insulin treatment on the lipid status of a certain patient. Or, on the other hand, why did the mentioned patients not get hypolipaemic drugs?
Response: Dear doctor, we understand and appreciate your concern. The patients involved in the study are low-income and receive care through the Unified Health System (SUS). Although the SUS offers free medical care, in Brazil, there is a shortage of health professionals specializing in treating DM1 in children and adolescents. CENID (Interdisciplinary Center for Diabetes), where the project was carried out, only started services in 2019.
This center was created precisely to meet the demand for specialized care for DM1 in the region. For this reason, most patients had inadequate treatment for DM1 and no guidance, monitoring or treatment for dyslipidemia. Thus, when patients enter CENID, most have already been diagnosed with DM1 for years but without adequate treatment.
Although at CENID, after the diagnosis of dyslipidemia, treatment was started, the outcome of the treatment is not part of the objective of this study, which prioritized the carrying out of a cross-sectional observational study based on the entry of patients into this treatment center.
It is worth noting that when patients entered this center for DM treatment, none were receiving treatment for dyslipidemia. Considering that from 2019 to 2020 the CENID outpatient clinic provided services only once a week and only in the morning, a long period of time was needed to collect data for the study.
Minor obstacles:
- The introduction must be improved with a re-selection of data about diabetes (too many data points are not linked to dyslipidemia).
Dear doctor, thank you for this comment. There is a restrict literature about dyslipidemia in Type 1 children. Please, see highlighted in yellow the modifications performed in this section (page 2, lines 66-74 and 82-85). We included some new references such as:
Zeng Q, Chen XJ, He YT, Ma ZM, Wu YX, Lin K. Body composition and metabolic syndrome in patients with type 1 diabetes. World J Diabetes. 2024 Jan 15;15(1):81-91. doi: 10.4239/wjd.v15.i1.81. PMID: 38313851; PMCID: PMC10835494.
Bezerra MF, Neves C, Neves JS, Carvalho D. Time in range and complications of diabetes: a cross-sectional analysis of patients with Type 1 diabetes. Diabetol Metab Syndr. 2023 Nov 27;15(1):244. doi: 10.1186/s13098-023-01219-2. PMID: 38008747; PMCID: PMC10680248.
Valerio G, Iafusco D, Zucchini S, Maffeis C; Study-Group on Diabetes of Italian Society of Pediatric Endocrinology and Diabetology (ISPED). Abdominal adiposity and cardiovascular risk factors in adolescents with type 1 diabetes. Diabetes Res Clin Pract. 2012 Jul;97(1):99-104. doi: 10.1016/j.diabres.2012.01.022. Epub 2012 Feb 13. PMID: 22336634.
- Table 1. Multiple doses of insulin are abbreviated as MDI, not as MID.
Dear doctor, the correction was performed in Table 1.
- Gentle English polishing is needed.
Response: Dear doctor, thank you for this suggestion. The manuscript was reviewed by a native.
Comments on the Quality of English Language: The manuscript was reviewed by a native.
- Table 1. Multiple doses of insulin are abbreviated as MDI, not as MID.
Response: The corrections was done.
Reviewer 2 Report
Comments and Suggestions for Authors
The authors (Melo et al., Investigating the Incidence of Dyslipidemia among Brazilian Children and Adolescents Diagnosed with Type 1 Diabetes Mellitus (T1DM) : A Cross-Sectional Study) proposed a cross-sectional study aiming to investigate in T1DM children and adolescents the prevalence of health outcomes (dislipidemia and diabetes with blood lipid biomarker s, HbA1c, fasting blood glucose,anthropologic data, exercise (physical activity level) and diet (kcal/kg).
Genral comments
The analysis used prevalence and logistic regression models and not odds ration, Why ?
Bootstrapping was used without explaination of the interest (resampling) .
Please mention the retrospective or prospective analysis and their associated biais.
Discuss the external validity, and the lack of a sensitive analysis.
The limitations should include the difficulty to derive causal relationships but is preliminary to cohort study analysis. Temporality as a limitation should be discussed.
For presentation Stobe statement as a quality factor should be presented in an annex.
What is the interest tu study Brasilian people for other scientists ?
Minor points.
Line 177 ANOVA
Line 200 Physical activity level, BMI
Table 3 TC and p values of 0.084 (NS) in the table but presented as significant in the results section.
Footnote table 3 for p-values 0.05, ApoA1
Figure 2 need a higher definition for readers
Table 5 please check the p values in the right colunm.
Line 345 incomplete sentence
Line 367 HbA1c values
Line 369 fructosamine ? please clarify
Statins are recomended despite LDL-C is not always the main factor, why this recommendation unstead PAL and diet prescription?
T1DM throughout the MS
A discussion for HDL-C and ApoA1 and the use of non-HDL is required
The reference section should be carefully revised : ex. ref. 31 35 36 42 etc.
Author Response
Response: Dear doctor, thank you for your time evaluating our manuscript. We performed the corrections according to your suggestions.
Genral comments
- The analysis used prevalence and logistic regression models and not odds ration, Why ?
Response: Dear doctor, thank you for this question. Prevalence data are already available in tables 2 and 4. In the study, we opted for linear regression analysis, instead of logistic regression, due to the fact that when using a quantitative variable in the outcome, greater sensitivity is observed to detect the effect of exposure factors, although it is not possible to make estimates of the odds ratio.
- Bootstrapping was used without explaination of the interest (resampling) .
Response: The use of the Bootstrapping technique to calculate the 95% confidence interval provides more precise estimates and the description of the technique used is described in the method in item 2.4. The Bootstrap technique involves estimating the confidence interval based on a sample of 1000 elements based on the percentile distribution of the data set. Although little used, there are already indications in the literature that this technique is safe and provides reliable estimates. Below is an excerpt from the method that describes the Botstrap technique. The 95%CI was calculated using the Bootstrap technique for percentile and resampling of 1000 sample elements.
- Please mention the retrospective or prospective analysis and their associated biais.
Response: Dear reviewer, considering that the study carried out was a cross-sectional observational study, there is a need to comment on bias associated with retrospective or prospective studies as these are longitudinal observational studies. Although the patients involved in the study received follow-up and treatment that will give rise to a prospective longitudinal observational study, the data are not yet available for analysis and presentation in this article.
- Discuss the external validity, and the lack of a sensitive analysis.
Response: We included the following paragraph in the Discussion (page 13, lines 399-406):
The use of all lipid parameters recommended for dyslipidemia screening contributes to increasing the external validity in relation to prevalence estimates. Many studies that carried out surveys of the prevalence of dyslipidemia in the population of children and adolescents with DM1 did not carry out estimates based on the dosage of all suggested lipid parameters and this could lead to prevalence estimates that are lower than reality. In Brazil, due to economic issues, the dosage of Apoliproteins A and B is not very frequent, which may be contributing to estimates of the prevalence of dyslipidemia lower than that observed in the present study.
- The limitations should include the difficulty to derive causal relationships but is preliminary to cohort study analysis. Temporality as a limitation should be discussed.
Response: Dear doctor, thank you for this suggestion. Please find the sentence, “Although the cross-sectional observational study has its limitations, regarding the cause and effect relationship, the results of the multiple linear regression analysis provide an important insight, both for directing clinical practice for the interdisciplinary care team, when constructing cohort studies more robust.” (Page 14, lines 420-424)
- For presentation Stobe statement as a quality factor should be presented in an annex.
Response: Dear doctor, thank you for this observation. The study followed Strobe recommendations for structuring and presenting a cross-sectional observational study (von Elm et al., 2007). Moreover, we include this information on page 4, lines 168-169.
- What is the interest tu study Brasilian people for other scientists ?
Response: In addition to the increase in the incidence of DM1 worldwide, the great diversity of the Brazilian population, in terms of genetic aspects, and accessibility to healthcare, allows the results of studies produced in Brazil to be used as a reference for studies in different locations
Minor points.
Response: Dear Doctor, thank you for your careful correction. We modified all the minor points along with the text.
Line 177 ANOVA. Response: Corrected.
Line 200 Physical activity level, BMI. Response: Corrected.
Table 3 TC and p values of 0.084 (NS) in the table but presented as significant in the results section. Response: Corrected.
Footnote table 3 for p-values 0.05, ApoA1. Response: Corrected.
Figure 2 need a higher definition for readers. Response: Corrected.
Table 5 please check the p values in the right colunm. Response: Corrected.
Line 345 incomplete sentence. Response: Corrected.
Line 367 HbA1c values. Response: Corrected.
Line 369 fructosamine ? please clarify.
Response: We removed fructosamine.
Statins are recomended despite LDL-C is not always the main factor, why this recommendation unstead PAL and diet prescription?
Response: We included (page 13-14, lines 415-419): Although physical exercise and diet are widely recommended for changes in lipid parameters other than LDL-c, in conditions where lifestyle modifications are not capable of producing significant improvement, the study of statins is recommended. It is worth highlighting that changes in lifestyle are part of the recommendations for the treatment of DM1, but an increase in the prevalence of a sedentary lifestyle among children and adolescents has been observed.
T1DM throughout the MS.
Response: We modified DM1 for T1DM along with all the manuscript.
A discussion for HDL-C and ApoA1 and the use of non-HDL is required.
Response: Included
The reference section should be carefully revised: ex. ref. 31 35 36 42 etc.
Response: The references were built with a reference manager.

Round 2
Reviewer 1 Report
Comments and Suggestions for Authors
The authors corrected the paper as suggested and explained the vague issues.
Reviewer 2 Report
Comments and Suggestions for Authors
The authors have modified and improved their manuscript.